# Complementary Methods to Improve the Depuration of Bivalves: A Review

**DOI:** 10.3390/foods9020129

**Published:** 2020-01-24

**Authors:** Antía Martinez-Albores, Aroa Lopez-Santamarina, José Antonio Rodriguez, Israel Samuel Ibarra, Alicia del Carmen Mondragón, Jose Manuel Miranda, Alexandre Lamas, Alberto Cepeda

**Affiliations:** 1Laboratorio de Higiene Inspección y Control de Alimentos. Departamento de Química Analítica, Nutrición y Bromatología, Universidade de Santiago de Compostela, 27002 Lugo, Spain; albores.antia@gmail.com (A.M.-A.); a_lo_san@hotmail.com (A.L.-S.); aliciamondragon@yahoo.com (A.d.C.M.); josemanuel.miranda@usc.es (J.M.M.); alexandre.lamas@usc.es (A.L.); 2Área Académica de Química, Universidad Autónoma del Estado de Hidalgo, Carretera Pachuca-Tulancingo Km 4.5, 42076 Pachuca, Hidalgo, Mexico; josear@uaeh.edu.mx (J.A.R.); israel_ibarra@uaeh.edu.mx (I.S.I.)

**Keywords:** depuration, bivalves, biotoxins, metallothioneins, chitosan, heavy metals

## Abstract

Bivalves are filter feeders that can accumulate and concentrate waterborne contaminants present in the water in which they live. Biotoxins, pathogenic bacteria, viruses, and heavy metals present in the aquaculture environment constitute the main hazards for human health. The most common method employed for combating waterborne pollutants in bivalves is depuration with purified seawater. Although this method is effective at increasing the microbiological quality of bivalves, in most cases, it is ineffective at eliminating other risks, such as, for example, viruses or heavy metals. Biological (bacteriocins and bacteriophages), physical (UV light, ozone, and gamma-irradiation), chemical (metallothioneins and chitosan), and other industrial processing methods have been found to be useful for eliminating some contaminants from seawater. The aim of this work was to provide a review of academic articles concerning the use of treatments complementary to conventional depuration, aiming to improve depuration process efficiency by reducing depuration times and decreasing the levels of the most difficult-to-erase contaminants. We conclude that there are different lab-tested strategies that can reduce depuration times and increase the food safety of bivalve produce, with possible short- and long-term industrial applications that could improve the competitivity of the aquaculture industry.

## 1. Introduction

The great challenge facing humanity in the coming decades is to secure food for the 9.8 billion people who are expected to inhabit the planet by around 2050 and 11.2 billion in 2100 [1]. In order to properly feed such a large population, it will be necessary to increase food production while respecting ecosystems and natural resources. The Food and Agriculture Organization of the United Nations states that aquaculture contributes to the efficient use of natural resources, food security, and economic development, with a limited and controllable impact on the environment [1]. Aquaculture is an activity that can contribute to obtaining higher yields of production by optimizing the breeding process. The world fish production reached, in 2016, a maximum of approximately 171 million tons, of which aquaculture represented 47% of the total [2]. Regarding European aquaculture, the most produced species in Europe are bivalves, such as mussels (*Mytilus* spp., including the species *M. edulis* and *M. galloprovincialis*), which represent more than 50% of the total European production [2].

A particularly important point from a food safety standpoint is that the feeding process of bivalve mollusks is by water filtration [3]. Via this process, they accumulate pathogenic bacteria, viruses, toxins, and chemical pollutants in their tissues that can pose a risk to public health [4,5]. These can then be transmitted to future consumers, with a high risk for public health [6,7].

To protect the consumer, aquaculture production areas are subject to programs of surveillance and control to avoid the presence of products carrying human pathogens [8]. The ideal procedure to obtain mollusks safely would be cultivation and harvesting in areas that are not subject to any type of contamination. However, this is unfeasible from a productive point of view because of the scarcity of these areas [7]. Consequently, depuration is a legal requirement in large countries for the marketing of fresh mollusks in order to protect consumers’ health.

Currently, the most widespread method to reduce contamination is depuration, with good results in the elimination of fecal bacteria, but with variable effectiveness in the elimination of other shellfish contaminants [9]. For this reason, the behavior and accumulation of specific contaminants by filtering organisms, together with the effectiveness of depuration for their elimination, is the subject of research studies that seek to obtain safe food to protect the final consumer.

Depuration processes usually exploit the natural filtering activity of bivalve mollusks, which results in the expulsion of intestinal contents. This process reduces the probability of the transmission of infection agents to consumers through the consumption of contaminated mollusks [10]. The conventional methods used in the mollusk depuration process are chlorine, ultraviolet (UV) light, and ozone [11]. Chlorine can affect the pumping activity of mollusks, cause organoleptic changes in their meat, and cause the presence of chlorinated metabolites, such as trihalomethanes, which have carcinogenic potential [11]. In order to avoid these inconveniences, depuration plants eliminate residual chlorine from mollusks by degassing with thiosulphate and vigorous agitation of the water before introducing them into depuration facilities [12].

The use of UV light for depuration also presents disadvantages, since its efficiency depends directly on the water turbidity and flow speed, which makes this method poorly applicable for large volumes [13]. Ozone also has more variable efficacy, although like chlorine it can form potentially cancerous derivatives, such as bromates [12].

Although conventional methods are routinely successful in eliminating bacterial agents, they are rarely or not at all effective in eliminating other health risks, such as viruses, toxins, and heavy metals [10]. As a consequence, there have been some outbreaks related to mollusk consumption caused by viruses, even when the mollusks were previously purified [14]. The most common viral pathogens involved in mollusk-caused outbreaks were norovirus (higher than 80%), followed by hepatitis A virus [15]. Oysters are the most frequently implicated shellfish in viral outbreaks [15]. This increased frequency is caused by the fact that the oysters are, in many cases, consumed raw, and that it was reported that in some cases noroviruses can bind specifically to carbohydrate structure antigens in the oyster gut and can be internalized within cells of both digestive and non-digestive tissues [16,17]. Such specific ligand interactions do not only serve to bioconcentrate noroviruses in mollusks as compared to their environment but also to anchor them during the depuration processes, thus rendering depuration, which is efficacious for bacterial elimination, insufficient for the elimination of specifically bound viral particles [17].

For this reason, it is necessary to develop complementary methods that, combined with conventional depuration methods, reduce the presence of different types of contaminants in bivalves. To the best of our knowledge, this is the first review focused on complementary depuration methods that are applicable to improve or extend the efficacy of depuration of live bivalves.

## 2. Modification of Marine Water Employed in Depuration

Studies have shown that a rise in water temperature can favor the purification process because of an increase in the pumping and enzymatic activity by mollusks [12,18]. However, there are limits regarding the optimum purification temperature specific to each species of mollusk [19]. Thus, for different mollusk species, there are water temperatures that, if they are exceeded, may cause negative effects in mollusks, such as a decrease in their feeding ingestion, absorption, and clearance rates, and even an increase in mollusk mortality [19]. Additionally, the use of hot marine water with different species, geographic areas, and even the season of the year showed contradictory results in terms of the elimination of both bacteria and viruses from bivalves [9,12]. In this sense, water at 25 °C showed a more effective depuration of both bacteria and viruses (poliovirus and hepatitis A virus) in oysters (*Crassostrea virginica*) compared to water depuration at 15 °C [9]. The same results were observed in clams (*Mercinaria mercinaria*) due to the more specific physiological characteristics of each species (Table 1). On the other hand, a posterior work did not find significant differences in the elimination of hepatitis A virus from mussels (*M. galloprovincialis*) depurated with marine water at 13 or 17 °C [13]. The inefficacy in reducing hepatitis A virus in mussels by increasing temperature could be explained by a negative influence of that increased temperature on the feeding processes and digestive activity of the mussels [20].

The disinfection of seawater is essential for the efficient purification of mollusks, especially in recirculation systems. One of the most used methods is disinfection by UV light, but the times and doses that are used regularly for bacterial depuration are not enough for the elimination of viruses. Thus, some enteric viruses, such as caliciviruses, show high resistance to UV treatment, requiring a UV light dose of approximately 40 mJ/cm^2^ for inactivation [8]. Another highly resistant family are the adenoviruses, which require UV light at doses higher than 170 mJ/cm^2^. While the degradation of the viral genome is achieved through the application of UV light on the virus, it also causes damage to the viral capsid or incapacitates the virus to infect cells [21]. Ultraviolet irradiation is effective in reducing noroviruses surrogated and hepatitis A viruses on the surface of the product but cannot inactivate viruses deep within shellfish [16]

It was shown that the amount of domoic acid (DA) produced by *Pseudo-nitzschia multiseries* and *P. australis* increases because of a defense mechanism against low levels of Fe^3+^ and Cu^2+^ that cause stress in these diatoms [22]. It was also proven that DA can be degraded by simple exposure to visible-spectrum UV light, and that this degradation is faster when Fe^3+^ is present in the seawater at up to 3 µM [18]. Therefore, in order to eliminate DA from the purification water, it is important to ensure that an adequate concentration of Fe^3+^ is dissolved in the water.

## 3. Depuration by Biological Methods

Bivalves can naturally contain compounds that show antibacterial, antiviral, antioxidant, and immunomodulatory effects [24]. Among them, both antimicrobial peptides and polysaccharides have received great attention in recent years [24,25,26].

Antimicrobial peptides are expressed by a range of animals as part of the primary defense system against pathogenic microorganisms. A large variety of small antimicrobial peptide families have been described from mollusks, mainly mussels (*Mylitus* spp.) [25]. Peptides have a small structure and provide a wide range of antimicrobial activities [27]. Antimicrobial peptides have been investigated because of their potential as new pharmaceutical substances, both for human and animal purposes [28]. In the context of intensifying aquaculture, antimicrobial peptides have been proposed as substitutes for antibiotics to prevent the selection of bacterial-resistant strains and reduce the environmental disadvantages of the use of antibiotics. However, only a few applications have been reported in extensive aquaculture, and none were aimed at being used in the depuration process [28]. Examples of works that have demonstrated the in vitro activity of antimicrobial peptides against mollusks pathogens include Defer et al. [25], whose isolated peptides from hemolymph bacteria showed inhibition in oysters (*Crassostrea gigas*) against *Bacillus megaterium* and *Micrococcus luteus,* or Ghorbanalizadeh et al. [27], whose isolated peptides from cockles (*Cerastoderma* spp.) showed inhibition against *Salmonella typhi*, *S. parathypi*, and *Staphylococcus aureus*.

In the same way, polysaccharides extracted from bivalves were shown to have a large variety of bioactivities and are usually employed in the prevention and treatment of a large variety of human diseases, including antibacterial or antiviral activities [24]. However, as in the case of antimicrobial peptides, their applications have been oriented towards other activities and not towards the purification process.

Another promising mechanism that presents interesting utilities is the use of probiotics. The term probiotics is related to bacterial species with beneficial characteristics or that can protect bivalves against infectious agents [6]. Probiotic bacteria can colonize the bivalve’s digestive gland and compete for space and nutrients with potentially pathogenic bacteria [10]. Other actions include the synthesis of antimicrobial compounds and digestive enzymes that improve food conversion and nutrients’ assimilation by the host and strengthen its immune system and capacity to tolerate stress [29]. The current usage of probiotics is today scarcer in mollusks than in other marine animals, such as shrimp and marine fish, but growing interest and recent advances in this field demonstrate their value [30,31]. Therefore, the application of beneficial bacteria in aquaculture can reduce or even eliminate the need for depuration of bivalves or improve the effectiveness of the technique (Table 2). In addition, these bacteria produce vitamins, enzymes, and/or essential fatty acids for the bivalve; stimulate its immune system; and can produce substances called bacteriocins with a broad spectrum of pathogen inhibitory activity [6]. Bacteriocins are proteinaceous molecules produced from bacterial strains from animals or the marine environment and have activity against other bacteria [6]. Their isolation of bacteriocin-producing bacteria was previously reported in a large variety of seafood and was applied mainly for the biocontrol of *Listeria monocytogenes* in processed foods [32]. In other cases, probiotics were able to improve the ability of mollusks to survive infection by pathogens, such as lion pan scallops (*Nodipecten subnodosus*) infected by *Vibrio alginolyticus* [30].

At the in vitro level, various bacteriocins produced by marine bacteria showed important activity against pathogenic marine bacteria. In this sense, several types of bacteria with important inhibition capacities against Gram-positive bacteria, such as *S. aureus*, *Bacillus subtillis*, or *Enterococcus faecium*, and even against yeast (*Candida albicans*), and molds (*Aspergillus niger* and *Fusarium oxysporum*) were isolated from the ark clam (*Anadara broughtoni*) [32]. Pinto et al. [33] isolated *E. faecium* and *Pediococcus pentosaecus* from oysters (*Ostrea edulis*) and clams (*Venerupis rhomboides*) with inhibition activity against Gram-positive bacteria, including *L. monocytogenes*, but no inhibition capacity against Gram-negative bacteria. Lee et al. [34] isolated *Lactobacillus rhamnosus* from oysters (*C. gigas*), which showed inhibition activity against *V. alginolyticus* and *V. proteolyticus*.

Bacteriocins can be employed in two different ways for the treatment of bivalves. The first consists of engulfing the bacteriocin-producing microorganism in the purification water at an adequate concentration to ensure that all the bivalves in the purifier meet the bacteria [6]. However, the contact with chlorinated, ozonized, or UV light-treated water that is used in conventional purification can also damage the bacterial species that produce the bacteriocins. To avoid this, the second option consists of the direct application of encapsulated bacteriocin. Although bacteriocin encapsulation is a relatively expensive process, it has already been applied in human and veterinary medicine [38]. Its combined use with the purification process was experimentally tested to reduce the amount of *V. parahaemolyticus* in oysters [35,37]. The results show a reduction of nearly 6 log CFU/g in *V. parahaemolyticus* in oysters (*O. plicatula*) after a combined purification process with phages for 72 h.

Another interesting way to improve the depuration process in mollusks is the use of bacteriophages, which were employed in cockles (*Cerastoderma edule*) and eliminated concentrations of about 5 log CFU/g of *Escherichia coli* after 4 h of depuration. These methods showed a great acceleration of the process with respect to conventional depuration methods. In addition, the phages employed in the process do not remain in the final product, since they are destroyed after a period of exposure to ultraviolet light from solar radiation, which improves the product’s safety [36].

## 4. Depuration by Physical Methods

Different physical methods tested at an experimental level showed effectiveness in the elimination or reduction of pathogenic agents in bivalves. Some examples are the application of various temperature treatments [39,40], X-ray irradiation [41], γ-irradiation [42,43], ozone [44,45,46,47], and the application of high hydrostatic pressure (HHP) alone [48] or combined with bacteriophages [48]. The effectiveness of the use of the combination between several of the previous techniques, such as the high hydrostatic pressures combined with a moderate heating [49], or the high hydrostatic pressures combined with the use of bactericidal phages [48], was also demonstrated (Table 3). However, the bivalves die during these processing techniques, modifying their nutritional and organoleptic characteristics; because of this, the application of these techniques is only useful on products that are going to be commercialized or transformed and never on fresh products [36].

Physical methods based on the application of special temperatures are aimed to decrease microbiological risks (*Vibrio* spp.) and can consist in flash freezing [39], water refrigeration [40], or thermal treatment combined to HHP [49]. Refrigeration is the usual method to maintain and transport live bivalves, and the use of slurry ice was previously demonstrated to improve their microbial and sensory quality [50].

Food irradiation is recognized as an effective technology for the elimination of pathogens that contaminate crude food [51,52], thus there are very few studies on the application of this treatment for the depuration of bivalves [53]. This treatment is only regulated in some EU countries, where the use of 3 kGy radiation is allowed for the treatment of fish and shellfish, which can be increased to 5 kGy for unpeeled and/or decapitated prawns [40]. In addition, the European Food Safety Authority (EFSA) [54] also intends to set a limit of 5 kGy for the irradiation of both fresh and frozen seafood products.

A previous work demonstrated that there is a significant reduction (10%–41%) in okadaic acid (OA) levels in mussels after the application of irradiation [42]. In addition, in their study, they also evaluated negative (non-toxic) samples to verify that there was no formation of acute toxicity-producing compounds; samples remained negative after treatment. However, they did not rule out the possibility that this treatment may form compounds that produce long-term toxicity or which have carcinogenic effects, such as 2-alkylcyclobutanones, which are radiolytic derivatives of triglycerides that are contained exclusively in irradiated foods and which have been experimentally shown to promote colon cancer [55].

Studies regarding the application of ozone to lipophilic toxins and diarrheic shellfish poisoning (DSP) toxins are limited [44,45]. Additionally, these studies only assess the effectiveness of ozone treatment on the water in which the mollusks are kept, not on the final product, since they can only be applied before harvesting while mollusks are alive.

To evaluate the efficacy of ozone treatment applied to the final product in the elimination of DSP toxins, Louppis et al. [42] ozonized samples of homogenized mussels and whole unshelled mussels inside a refrigerator at 4 °C at a dose of 15 mg/kg for 6 h. The results obtained show that this treatment is effective in reducing the amount of OA and its derivatives independent of the initial degree of contamination. Thus, in the worst case, this treatment provided a 21% reduction in the content of this toxin in mussels (*M. galloprovincialis*), which experimentally was enough to change a product that could not be marketed to one that complies with legal limits. Thus, it should be noted that a greater reduction in OA was obtained in homogenized mussel samples than in whole mussel samples. This was attributed to the fact that the treated mussel tissue was from the hepatopancreas, which has a much higher lipid content than whole mussel tissue, a factor that interferes with the capacity of ozone to interact with the toxin, because OA is lipophilic [42].

A possible explanation for the effectiveness of treatment with ozone on the OA content of mussels could be the interaction of the gas with the double bonds of the OA molecule [56]. It is known that ozone attacks the double bonds of organic compounds [57] and that the OA molecule contains several double bonds that could be a potential target for ozone. Therefore, the alteration in the OA structure that occurs after the treatment could be caused by the reduction in its concentration, because the detection procedures would not recognize the altered molecule. Consequently, it is necessary to continue investigating the mechanism of action of ozone to confirm its effectiveness and potential applications in industry.

## 5. Depuration by Chemical Methods

Conventional purification processes show slow elimination of certain undesirable compounds, such as marine toxins and heavy metals. Bivalves accumulate heavy metals slowly throughout their life, and their elimination is tremendously slow when treated only with clean seawater [58]. The heaviest heavy metal traditionally known to be dangerous to human health is Hg^2+^, with greater importance in pregnant women and children [59,60]. However, other targets, such as Cd^2+^, also represent an important risk, as a consequence of their solubility and because they are capable of producing problems in the immune and reproductive systems and have a potentially teratogenic effect [61]. The fact that this element is more soluble in acidic media implies that its presence in fishery products will be greater in the future because the oceans are undergoing an acidification process as a consequence of climate change [61].

In addition to fighting microbiological risks by co-purification, it is necessary to develop systems capable of reducing chemical agents in bivalves. One of the strategies proposed for this purpose is the use of chelating agents, which join heavy metals or other toxins, reducing their availability or facilitating their elimination [59]. Chelating agents currently present great potential for use in the food and health fields due to their antimicrobial, immunomodulatory, antitumor, and antioxidant effects, as well as their capacity to chelate and diminish heavy metal availability [62]. The specific usages of chelating agents in bivalves depuration can be shown in Table 4.

Among the chelating substances of interest for their application in aquaculture, metallothioneins (MTs), which are naturally produced by bivalves [58], are highlighted. MTs are proteins rich in cysteine and have a low molecular weight. They cover a wide range of organisms and show a remarkable affinity for targets, such as Zn^2+^, Cd^2+^, and Ca^2+^ [62,63]. The MTs contained in bivalves, such as mussels (*M. edulis*), play an important role in the transport and storage of heavy metals, but they also provide them with a certain protective function (detoxification effect) against excessive amounts of non-essential metals, such as in the case of Cd^2+^, Ag^2+^, or Hg^2+^ [58].

Recent work showed that the addition of MTs obtained from fishing subproducts achieved a reduction, through the formation of complexes containing Cd^2+^ in mussels (*M. edulis*) in the gills, mantle, and viscera, of around 30%, 40%, and 25%, respectively, after a 15-day exposure [64]. The same authors in a further work demonstrated a better elimination (about 50%) of Cd^2+^ from mussels (*M. edulis*) when employing MTs protein hydrolysate complexed to Fe^2+^ [64].

In a previous study, it was demonstrated that using MTs at the same time as traditional methods of depuration failed in reducing the concentration of Cd^2+^ significantly because its depuration kinetics are extremely slow [65]. Baudrimont et al. [66], in a previous work, also obtained good depuration results for Cd^2+^, Al^2+^, and Hg^2+^ using MTs in Asian clams (*Corbicula fluminea*).

Other natural compounds, such as chitosan, can also achieve this chelating action of heavy metals. Chitosan is a long-chain polysaccharide obtained by the distillation of chitin from crustacean shells [67]. More recently, Widiah Ningrum et al. [68] developed a system to reduce the level of Hg^2+^ in green mussels (*Perna viridis* L.) and in cockles (*Anadara granosa* L.). Chitosan is only soluble under acid conditions; for this reason, it is essential to use it in a way that favors its solubility [67]. Thus, chitosan can be dissolved in 5% acetic acid to form a gel. This gel with chitosan is administered by pumping it, together with ozonized seawater (at a concentration of 1.5 mg/L ozone and 0.5 mg/L chitosan), at a rate of 1.3 m/s for 1 day. This achieved a reduction in the Hg^2+^ content of more than 90% in mussels (*Perna viridis* L.), and of around 85% in the case of cockles (*A. granosa* L.) [68].

In another work [69], chitosan was also used for the experimental purification of oysters (*Ostrea rivularis*) over 7 days. During this period, it was possible to reduce the quantity of paralytic shellfish-poisoning (PSP) toxins in oysters by more than 60% when using chitosan, and by more than 85% when chitosan was administered in combination with *Chlorella* microalgae.

In the same way, Huang et al. [67] employed Chinese scallops (*Chlamys ferrari*) to study the purification process over 7 days with different concentrations of chitosan combined with calcium to facilitate its solubility. These researchers measured the concentrations of Cd^2+^, Ca^2+^, and Zn^2+^, demonstrating that, by means of this system, the concentrations of Cd^2+^ were reduced by 18% while those of Ca^2+^ and Zn^2+^ were not reduced by significant amounts. This uncontrolled reduction of Cd^2+^ implies an increase in the purification speed with respect to conventional methods [67].

## 6. Methodologies for Decontamination during Industrial Processing

Contamination by DSP toxins in mussels and by amnesic shellfish poisoning (ASP) in scallops are very important concerns in European aquaculture. When the mussels are contaminated with OA, natural detoxification takes several weeks; after that, the phenomenon of proliferation of toxic algae occurs [70], which causes great economic losses in the aquaculture sector. The problem is greater in the case of scallops (*Pecten maximus*), because their metabolism produces extremely slow detoxification processes and they can remain toxic for several months [71].

Several treatments have been studied to evaluate the reduction of toxicity by ASP in scallops, individually and in combination, including evisceration (extirpation of the digestive system and hepatopancreas), thermal treatment carried out on the product after removing the meat from the shell, applying a series of cooking and washing methods that finished off each sterilization in an autoclave at 116 °C for 54 min, and freezing to −20 °C. All methods resulted in an insignificant reduction in toxicity; the only method that was able to reduce the values to the legal limit was ablation of the hepatopancreas, achieving the almost complete elimination of the toxin [45].

The application of a thermal process was able to reduce the levels of PSP toxin significantly, especially in mussels (*M. edulis*), and even in clams (*Rudipates decussatus*) and cockles (*Cerastoderma edule*). The process consisted of a succession of washing and heating phases; the maximum peak temperature was 98 °C for 9 min followed by autoclaving at 116 °C for 54 min [45]. At the experimental level, it was also proven that the use of alkaline solutions followed by cooking and washing reduces the levels of this toxin [45].

Regarding DSP toxicity in mussels (*M. edulis*), the effect of freezing, ozonization, thermal treatment, and thermal treatment with additives was analyzed; diverse results were obtained for the elimination of OA. Detoxification for a month or more did not manage to modify the levels of OA [45]. There are references to the possibility of reducing its levels by using supercritical CO_2_, but the product obtained using this technique is not acceptable from a commercial point of view [72]. Another mechanism studied is the addition of n-acetylcysteine, a precursor of glutathione that would increase the speed of the detoxification mechanisms [73].

The maintenance of mussels (*M. edulis*) in ozonized sea water with a redox potential >450 mV for 24 h was not able to eliminate the DSP toxin from its interior, but it was able to cause a significant increase in the proportion of OA/total DSP [45], which shows that the oxidative power of ozone changes the profile of toxins in mussels.

Thermal treatment of mussels is not effective for decreasing the content of DSP toxin. However, the content of ASP and PSP toxins reduced significantly but without reaching values under the legal limits [45]. To achieve effective decontamination, it is necessary to combine thermal treatment with other methods, such as evisceration and freezing for ASP, and freezing for PSP.

There are several hypotheses regarding the difficulty of eliminating DSP toxin; in the scientific literature, there are references to OA being stored in different compartments within the organism presenting different isoforms [45]. Thus, OA shows different kinetics of detoxification and interconnection between its isoforms over time [74]. Furthermore, it has also been proposed that within these compartments, the toxins remain stored in inactive liposomes [75], which complicates the access to treatments, even though the main place of storage is the hepatopancreas.

## 7. Conclusions

There is no single method that can be successfully applied to all species of bivalves and protects against all sanitary risks. Some of the mentioned methods showed promising results at the experimental level, even for the elimination of very persistent pollutants. However, they have not yet been tested at the commercial level. For this reason, it is fundamental to transfer the knowledge to the production sector by promoting links between research and industry. The final aim is to improve the competitiveness of bivalve aquaculture, a sector of great current importance and which will be even more important in the future. In addition, these new methods in combination with traditional purification results in an increase in food security for the population.

## Figures and Tables

**Table 1 foods-09-00129-t001:** Treatments applicable to depuration water.

Reference	Treatment	Dosage and Time	Depuration Against	Efficacy
[23]	UV light + Fe^3+^	Continued exposure of light (full spectrum light and UV) + Fe^3+^ 0.3 mM for 22 h	Domoic acid	Degradation of 41% in the better results
[8]	UV light	Continue exposure at 44 mJ/cm^2^ for 24 h	Adenovirus and norovirus	99.9% of elimination of adenovirus and norovirus after 24 h
[9]	Combinations of temperature, salinity, turbidity, pH, and the presence of algae (*Isochrysis*)	Continue exposure for 5 days	*Escherichia coli*, *Enterococcus faecalis*, coliphage MS2, Poliovirustype-1 and Hepatitis A virus	In both clams (*Crassostrea virginica*) and oysters (*Mercinaria mercinaria*), bacterial indicators were depurated faster than viral indicators

**Table 2 foods-09-00129-t002:** Bacterial species producing bacteriocins isolated from bivalves.

Reference	Bacterial Species	Origin	Inhibition Against	Results
[34]	*Lactobacillus rhamnosus*	Oysters (*Crassostrea gigas*)	In vitro agar test of inhibition against pathogens	Good inhibition against *Vibrio alginolyticus* and *V. proteolyticus* and poor inhibition against *Edwardsinella tarda*
[6]	*Enterococcus hirae*	Mussels (*Mytilus galloprovincilais*)	In vitro agar test of inhibition against bacterial pathogens and on cellular lines for viruses	Good antibacterial activity against *Listeria monocytogenes,* and *Enterococcus faecalis.* Low antibacterial activity against *L. innocua,* good antiviral activity against Hepatitis A virus and Norovirus
[35]	Bacteriophagues	Oysters (*Crassostrea gigas*)	*V. parahaemolyticus*	Bacterial growth inhibition from 1.4 × 10^6^ CFU/mL to 1.4 × 10 CFU/mL
[36]	Bacteriophagues	Cockles (*Cerastoderma edule*)	*Escherichia coli*	Reducing *E. coli* counts about 5 log CFU/g after 4-h period of depuration
[33]	*E. faecium* and *Pediococcus pentosaecus*	Oyster *(Ostrea edulis)* and clams *(Venerupis rhomboides)*	In vitro agar test of inhibition against pathogen and spoilage bacteria	Inhibition against Gram-positive bacteria, such as *L. monocytogenes, L. innocua, Staphylococcus aureus*, or *Bacillus cereus.* No inhibition against Gram-negative bacteria
[32]	Various bacterial species from genera *Bacillus, Paenibacillus, Saccharorhix, Pseudomonas* and *Sphingomonas*	Ark clams *(Anadara broughtoni)*	In vitro agar test of inhibition against bacteria and in vitro agar modified method for fungi and yeast	Inhibition activity of various strains isolated against Gram-positive bacteria, such as *S. aureus, B. subtillis*, and *E. faecium*, and even against yeast (*Candida albicans*) and molds (*Aspergillus niger* and *Fusarium oxysporum*)
[37]	Bacteriophagues	Oysters (*O. plicatula*)	*V. parahaemolyticus*	Depuration at 16 °C with bacteriophage decreased *V. parahaemolyticus* in oysters, by 2.35–2.76 log CFU/g within 36 h
[30]	*Bacillus* and *Lactobacillus* mix	Lion paw scallops (*Nodipecten subnodosus*)	*V. alginolyticus*	Increase in survival of juveniles of catarina scallop (*Argopecten ventricosus*) in 120 h
[31]	*Enterococcus faecium*	Clams (*Tapes decussatus*)	*L. monocytogenes*	In vitro inhibition activity
[25]	Peptides isolated from hemolymph bacteria (not identified)	Oysters (*C. gigas*)	*Bacillus megaterium* and *Micrococcus luteus*	In vitro inhibition activity
[27]	Antimicrobial peptides	Cockles (*Cerastoderma* spp.)	*Salmonella typhi, S. paratyphi* and *S. aureus*	In vitro inhibition activity for both ethanolic and methanolic solutions against *Salmonella* and *S. aureus*

CFU: Colony form units.

**Table 3 foods-09-00129-t003:** Complementary depuration methods using physical procedures.

Reference	Treatment	Dosage and Time	Bivalve Species	Inhibition Against (Efficacy)
[48]	High hydrostatic pressure (HHP)	550 MPa for 5 min	Blue mussels (*Mytilus edulis*)	*Shigella flexneri* and *Vibrio cholerae* (complete elimination from 3.8 log CFU/g)
[44]	Ozonation	360 mg ozone/h for 3 days	Mussels (*M. galloprovincialis*)	Diarrheic shellfish poisoning (DSP) reduced toxicity in mouse after 3 days
[39]	Flash freezing and frozen	Flash freezing, followed by storage at −21 +/−2 °C for 5 months	Pacific oysters (*Crassostrea gigas*)	*Vibrio parahaemolyticus* and *Vibrio vulnificus* (3.52-fold log MPN/g)
[42]	Ozonation	15 mg/kg for 6 h	Mussels (*M. galloprovincialis*)	Okadaic acid (21%–66% reduction)
[42]	γ-irradiation	6 kGy	Mussels (*M. galloprovincialis*)	Okadaic acid (10%–41% reduction)
[41]	X-Ray	1–5 kGy	Oysters (*Crassostrea virginica*)	*V. parahaemolyticus* (4-fold log CFU/g)
[43]	γ-irradiation	6, 12, and 24 kGy	Mussels (*M. edulis*)	Domoic acid (40%–100%), azaspirazids (15%–50%), Okadaic acid (0%–30%), pectenotoxin (30%–75%), yesotoxins (0%–15%), depending of the dose
[40]	Refrigeration	Depuration at controlled temperature between 7–15 °C for 5 days	Oysters (*C. gigas*)	*V. parahaemolyticus* (3-fold log MPN/g)
[46]	Ozonation under different pH	1.24 V	Chemical analyses and mice bioassay	Ozone was more effective under acidic conditions and combined with hydrogen peroxide than alone conditions (2.07 V)
[47]	Ozonation	25 mg ozone/L for 30 seg	HPLC and fish (*Cyprinodon variegatus*) bioassay	*Gymnodinium breve* toxins showed 3-log CFU cycle reduction in the total toxin recovered after 10 min (135 mg/L) of ozone exposure
[49]	Temperature combined by high hydrostatic pressure	HHP at ≥275 MPa for 2 min followed by heat treatment at 45 °C for 20 min; HHP at ≥200 Mpa for 2 min followed by heat treatment at 50 °C for 15 min	Oysters (*C. virginica*)	*V. parahaemolyticus* and *V. vulnificus* (3-fold log MPN/g)

CFU: Colony form units; HHP: high hydrostatic pressure; Mpa: Megapascal; MPN: Most probable number.

**Table 4 foods-09-00129-t004:** Chelating agents used in bivalve chemical hazards’ depuration.

Reference	Chelating Agent	Dosage and Time	Bivalve Species	Inhibition Against (Efficacy)
[66]	Metallothioenins (MTs)	ND	Asiatic clams (*Corbicula fluminea*)	Cd^2+^ sequestered by the MTs fraction represented 40% of the total Cd^2+^ bioaccumulated in the soft body of the mollusks, compared with 4%–9% for total accumulated Zn^2+^
[67]	Chitosan oligosaccharide + Ca^2+^ (COS-Ca)	Different doses ranging 1.75–8.75 mg/L for 6 days	Scallops (*Chlamys**Ferrari*)	COS-Ca reduced Cd^2+^ content of the scallops, with highest depuration rate (47%) observed on day 3. Additionally, increased Ca^2+^ content (73.9%) on day 6, and did not significantly affected Zn^2+^ content
[69]	Chitosan, *Chlorella* and Chitosan + *Chlorella*	8 × 10^3^ cells/mL *Chlorella*, 0.05 g/L chitosan, and combination of both	Oysters (*Ostrea rivularis*)	Toxicity caused by paralytic shellfish poisoning decreased from 9.07 mouse units (MUs) to 1.41 MUs using chitosan and 0.12 mouse units using chitosan plus *Chlorella*
[58]	MTs (protein hydrolysate-Fe^2+)^	40 mg/L protein hydrolysate-Fe^2+^ for 15 days	Blue mussels (*Mytilus edulis*)	Cd^2+^ concentration in blue mussel decreased from 46.1 to 23.3 µg/g
[58]	MTs (hydrolysis peptide–metal element complexes (Fe^2+^, Zn^2+^, Ca^2+^, or Hg^2+^)	Different concentrations of MTs (5, 10, 15, and 20 mg/L) for 8 days	Blue mussels (*M. edulis*)	Cd^2+^ decreased in the range 25%–40% after exposure to 20 mg/L of hydrolysis peptide–metal element complexed to Fe^2+^, Zn^2+^, and Ca^2+^ No significant decrease was found for hydrolysis peptide–metal element complexed to Hg^2+^
[68]	Combinations between chitosan, ozone and hydrodynamic treatment	1.5 mg/L ozone, 0.5 mg/L chitosan and 1.3 m/s hydrodynamic treatment for 60 min	Green mussels (*Perna Viridis* L.) and blood cockles (*Anadara granosa* L.)	The most effective combination was chitosan-ozone, achieving a Hg^2+^ depuration of 96.5% in green mussels and 87% in blood cockles

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
