# Peer review of "Complementary Methods to Improve the Depuration of Bivalves: A Review"

_foods, 2020, doi:10.3390/foods9020129_

Round 1

Reviewer 1 Report

The manuscript contains some information regarding bivalve depuration, but it is not well written in general. The content of the manuscript is moderate, and it has to be enriched (add more refs), be more comprehensive and clearer. The scientific expressions are too complicated in most cases, therefore they have to be rephrased using a simple and clearer way. English is not understandable in many cases. There is a plethora of typographical and grammatical errors throughout the text making the article not good to convey the scientific meaning correctly. English has to be edited by a native user with knowledge on this scientific field.

Abstract describes the essential information of the manuscript. However, English has to be edited throughout the abstract. Some corrections are suggested thereunder  

L 14-16 remove ‘Among these contaminants’ and rephrase the sentence e.g. ‘The most significant contaminants ….’ or ‘Biotoxins, pathogenic bacteria, viruses… constitute the main hazards … human health…in aquaculture environment’.

L 18 rephrase and be clear. What authors mean ‘other risk’?

L 19 remove ‘Therefore’ and rephrase e.g. ‘Biological (…), physical…methodologies have been found… effective… for decontamination of seawater’

L 25 change ‘Results show that’ to e.g. ‘Concluding, there are different lab…’

Introduction

English has to be edited

L 33 According to the Department of Economic and Social Affairs of FAO, the world population projected to reach 9.8 billion in 2050 (and 11.2 billion in 2100). Please search and replace

L 32-34 remove ‘in addition to obtaining energy’ and state clearly the great challenge that we face (to secure food for … billion people … by 2050).

L 34 This is not an ‘objective’. Please find a more appropriate expression to characterize it, taking into account the reports of the FAO or the World Bank.

L 32-43 the paragraph has be rephrased to give the important statements clearer. The expressions are too complicated in some cases.  

L 44-50 same as above. Also, please remove ‘In order to guarantee the safety…’ we cannot ‘’guarantee’’ food safety.

L 51 ‘pollution’ is often used for chemicals, plastics etc. ‘contaminants’ could be more suitable

L 53-56 rephrase and be clearer

L 57-61 rephrase

L 62-64 rephrase

L 75-83 rephrase

Section 2: which is the subject in this section? There are some info regarding the depuration of bivalves and some for the decontamination of the water. Authors have to search literature in depth for each field (depuration of bivalves and/or decontamination of the water) in order to present a more comprehensive section. There is a plethora of publications on the above fields, mainly on the decontamination of the water. Also, they must include the info in the table 1. In addition, English has to be edited.

L 85 ‘water temperature’ or ‘sea surface temperature’?? check refs and act accordingly

L 87 ‘there are limits’ which are those? clarify

L 89 ‘shows…’ there is not a link with the sentence in L 86-88. Please link

L 90 ‘better elimination’? Do authors mean ‘the depuration was more effective at 25oC compared to 15 for oysters’??

L 90-93 too long sentence to be understandable. Shorten and make it clearer

L 94-95 (temperature can favor the purification process) and L 85-86 (did not find significant differences in the elimination of …) are followed by the same ref (13). These statements confuse the reader. How we can state that the temperature can favor the purification process while significant differences in the elimination of pathogens were not observed? Please be clearer

L 96 rephrase  

L 96-98 which result? this has to be clearer

L 99-122 see the general comments above (section 2)

Table 1: the title and content are poor (see the comments above)

Section 3 English has to be edited, sentences be rephrased and more info be added

L 126-128 add refs regarding ‘These microorganisms can favor the growth and vitality of bivalves’

L 137-139 ‘it were isolated’ please remove and rephrase

Table 2 has to be enriched with more refs

L 164-165 too many words, please make it simple

L 221-222 rephrase and remove the ‘above all’

L 223 Bivalves accumulate heavy metals...

L 232-233 rephrase

Sections 5 – 6 same as above (check English, rephrase in most cases, check for more info, etc)

Conclusions

L 321 remove ‘As final conclusion’

The section has to be rephrased too.

Author Response

Reviewer 1

General comments:

The manuscript contains some information regarding bivalve depuration, but it is not well written in general. The content of the manuscript is moderate, and it has to be enriched (add more refs), be more comprehensive and clearer. The scientific expressions are too complicated in most cases; therefore they have to be rephrased using a simple and clearer way. English is not understandable in many cases. There is a plethora of typographical and grammatical errors throughout the text making the article not good to convey the scientific meaning correctly. English has to be edited by a native user with knowledge on this scientific field.

Response from authors: The authors want to thank the constructive comments from the reviewer. In fact, in none of the authors of the manuscript in a native English speaker and consequently it is normal that some English grammar mistakes are present. To avoid this problem, the authors used the MDPI English editing service, that corrected the English spelling in the revised version of the manuscript.

With respect to the need of adding more references, the original version was aimed to include mostly recently published articles, and there are no there are not many on this subject that have been published in recent years. Nevertheless, the authors followed the reviewer comments and in the revised version of the manuscript it was included 14 new references, in concrete:

Hodgson, K.R.; Torok, V.A.; Turnbull, A.R. Bacteriophages as enteric viral infections in bivalve mollusk management. Food Microbiol. 2017, 65, 284-293. Naïma, B.M.H. Human enteric viruses in bivalve molluscs: Contamination and detection. J. Sci. Technol. 2015, 4, 6. Razafimahefa, R.M.; Ludwig-Begall, L.F.; Thiry, E. Cockles and mussels, alive, alive, oh*-The role of bivalve molluscs as transmission vehicles for human norovirus infections. Emerg. Dis. 2019, 1, 17. Richards, G.P., McLeod, C., Le Guyader, F.S. Processing strategies to inactivate viruses in shellfish. Food Environ. Virol. 2010, 2, 183-193. Lee, R., Lovatelli, A., Ababouch, L., Bivalve depuration: fundamental and practical aspects. Food Agric. Organ. U. N. 2008, 1-139. Wang, L.C.; Di, L.Q.; Li, J.S.; Cheng, J.M.; Wu, H. Elaboration in type, primary structure and bioactivity of polysaccharides derived from mollusks. Rev. Food Sci. Nutr. 2019, 7, 1091-1114. Defer, D.; Desriac, F.; henry, J.; Bourgougnon, N.; Baudy-Floch, M.; Brillet, B.; Le Chevalier, P.; Fleury, Y. Antimicrobial peptides in oyster hemolymph: The bacterial connection. Fish Shellfish Immunol. 2013, 34, 1439-1447. Leoni, G.; De Poli, A.; Mardirossian, M.; Gambato, S.; Florian, F.;Venier, P.; Wilson, D.N.; Tossi, A.; Pallavicini, A.; Gerdol, M. Myticalins: A novel multigenic family of linear, cationic antimicrobial peptides from marine mussels (Mytilus) Mar. Drugs 2017, 15, 261. Ghorbanalizadeh, A.; Moshfegh, A.; Setorki, M. Evaluation of antimicrobial activity of peptides isolated from Cerastoderma and Didacta bivalves habitat in the southern shores of the Caspian Sea. J. Aquat. Anim. Health 2018, 4(1), 1-12. Destoumieux-Garzón, D.; Costa, R.D.; Schmitt, P.; Barreto, C.; Vidal-Dupiol, J.; Mitta, G. Antimicrobial peptides in marine invertebrate health and disease. Trans. R. Lond. B. Biol. Sci. 2016, 371(1695), 20150300. Campa-Córdova, A.I.; González, H.; Luna-González, A.; Mazón-Suástegui, J.M.; Ascencio, F. Growth, survival, and superoxide dismutase activity in juveline Crassostrea corteziensis (Hertlein, 1951) treated with probiotics. Hidrobiologica, 2009, 19, 151-157. Abasolo-Pacheco, F.; Campa-Códova, A.I., Mazón-Suástegui, J.M.; Tovar-Ramirez, D.; Araya, R.; Saucedo, P.E. Enhancing growth and resistance to Vibrio alginolyticus disease in Catarina scallop (Arcopecten ventricosus) with Bacillus and Lactobacillus probiotic strains during early development. Res. 2017, 48, 4597-4607. Lim, H.J.; Kapareiko, D.; Schott, E.J.; Hanif, A.; Wikfors, G.H. Isolation and evaluation of new probiotic bacteria for use in shellfish hatcheries: Isolation and screening for bioactivity. Shellfish Res. 2011, 30, 609-615 Sánchez-Valenzuela, A.; Benomar, N.; Abriouel, H.; Cañamero, M.M.; Gálvez, A. Isolation and identification of Enterococcus faecium from seafoods: Antimicrobial resistance and production of bacteriocin-like substances. Food Microbiol. 2010, 27, 955-961.

With respect to comments about “Abstract describes the essential information of the manuscript. However, English has to be edited throughout the abstract. Some corrections are suggested thereunder”.

According to the suggestions from the Reviewer, abstract section was corrected in the English spelling was corrected.

With respect to the comments about “L 14-16 remove ‘Among these contaminants’ and rephrase the sentence e.g. ‘The most significant contaminants ….’ or ‘Biotoxins, pathogenic bacteria, viruses… constitute the main hazards … human health…in aquaculture environment’.”

According to the suggestions from the Reviewer, the sentence was rephrased.

With respect to the comments about “L 18 rephrase and be clear. What authors mean ‘other risk’?”

According to the suggestions from the Reviewer, the sentence was rephrased. In this case, the term “other risk” means “risks other than microbiological”. To clarify it, it was added “as for example viruses or heavy metals.”

With respect to the comments about “L 19 remove ‘Therefore’ and rephrase e.g. ‘Biological (…), physical…methodologies have been found… effective… for decontamination of seawater’”

According to the suggestions from the Reviewer, the term “Therefore” was deleted, and the phrase was changed.

With respect to the comments about “L 25 change ‘Results show that’ to e.g. ‘Concluding, there are different lab…’”

The sentence was modified accordingly with the suggestion from the Reviewer

With respect to the comments about “Introduction, English has to be edited”

According to the suggestions from the Reviewer, English from Introduction as well as all manuscript was Edited by MDPI official editor.

With respect to the comments about “L 33 According to the Department of Economic and Social Affairs of FAO, the world population projected to reach 9.8 billion in 2050 (and 11.2 billion in 2100). Please search and replace”

According to the suggestions from the Reviewer, the information was updated, and the new reference was added in the text.

With respect to the comments about “L 32-34 remove ‘in addition to obtaining energy’ and state clearly the great challenge that we face (to secure food for … billion people … by 2050).”

According to the suggestions from the Reviewer, the sentence was modified.

With respect to the comments about “L 34 This is not an ‘objective’. Please find a more appropriate expression to characterize it, taking into account the reports of the FAO or the World Bank.”

According to the suggestions from the Reviewer, the sentence was modified, and was changed to “In order to properly feed such a large population”

With respect to the comments about “L 32-43 the paragraph has be rephrased to give the important statements clearer. The expressions are too complicated in some cases. “

According to the suggestions from the Reviewer, the paragraph was rewritten.

With respect to the comments about “L 44-50 same as above. Also, please remove ‘In order to guarantee the safety…’ we cannot ‘’guarantee’’ food safety.”

According to the suggestions from the Reviewer, the paragraph was rewritten.

With respect to the comments about “L 51 ‘pollution’ is often used for chemicals, plastics etc. ‘contaminants’ could be more suitable.”

According to the suggestions from the Reviewer, “pollution” was changes to “contamination”.

With respect to the comments about “L 53-56 rephrase and be clearer; L 57-61 rephrase; L 62-64 rephrase; L 75-83 rephrase”

According to the suggestions from the Reviewer, the four paragraphs cited were rewritten

With respect to the comments about “Section 2: which is the subject in this section? There are some info regarding the depuration of bivalves and some for the decontamination of the water.

According to the suggestions of the Reviewer, the method aimed to decontaminate water whose did not reported results in bivalves were deleted from this section. Only was maintained methods whose reported decontamination in bivalves.

With respect to the comments about “Authors have to search literature in depth for each field (depuration of bivalves and/or decontamination of the water) in order to present a more comprehensive section. There is a plethora of publications on the above fields, mainly on the decontamination of the water. Also, they must include the info in the table 1. In addition, English has to be edited.

According to the suggestions from the Reviewer, work regarding disinfections in water samples were deleted. The reviewer is right in that there are a plethora of articles describing disinfection methods in marine water, and consequently, to include all them would do the article too long and hard to read.

With respect to the comments about “L 85 ‘water temperature’ or ‘sea surface temperature’?? check refs and act accordingly”

According to the suggestions from the Reviewer, the reference was revised and, in all text, Polo et al. (2014) referred as “water temperature” and not “sea surface temperature” to refer the different ºc of the water employed in the depuration process. Thus, we consider maintaining the original term “water temperature”.

With respect to the comments about “L 87 ‘there are limits’ which are those? Clarify”

Thank you for your comment. The limits referred are the water temperatures that can damage biological activities and even cause mortality in each mollusk specie. Obviously, these limits are different for each mollusk specie, geographical region, etc., and consequently are too extensive to be specifically cited. To clarity it, it was included a new paragraph in the revised version of the text, with a new reference that can be consulted to the readers:

“Thus, for different mollusks species there are a water temperature that if it if exceeded, may cause negative effects in mollusks such as a decrease in their feeding ingestion, absorption and clearance rates and even an increase in mollusks mortality [Lee et al., 2008, 12]”.

Reference:

Lee, R., Lovatelli, A., Ababouch, L., Bivalve depuration: fundamental and practical aspects. Food Agric. Organ. U. N. 2008, 1-139.

With respect to the comments about “L 89 ‘shows…’ there is not a link with the sentence in L 86-88. Please link”

According to the suggestions from the Reviewer, a link was added.

With respect to the comments about “L 90 ‘better elimination’? Do authors mean ‘the depuration was more effective at 25oC compared to 15 for oysters’??”

According to the suggestions from the Reviewer, the phrase was corrected

With respect to the comments about “L 90-93 too long sentence to be understandable. Shorten and make it clearer”

According to the suggestions from the Reviewer, the phrase was corrected

With respect to the comments about “L 94-95 (temperature can favor the purification process) and L 85-86 (did not find significant differences in the elimination of …) are followed by the same ref (13). These statements confuse the reader. How we can state that the temperature can favor the purification process while significant differences in the elimination of pathogens were not observed? Please be clearer”

Thank you for your comment. In the cited text, information contained in lines 85-86 are referred to a general statement, that should have been attributed to the reference number 12 (Lees et al., 2010) and not for number 13 (Polo et al., 2014). Information stated in lines 94-95 are referred to a specific result of the work cited as reference number 13 (Polo et al., 2014). In was changed in the revised version of the manuscript accordingly.

With respect to the comments about “L 96 rephrase. L 96-98 which result? this has to be clearer”

According to the suggestions from the Reviewer, the phrase was corrected

With respect to the comments about “L 99-122 see the general comments above (section 2)”

According to the suggestions from the Reviewer, the phrase was corrected

With respect to the comments about “Table 1: the title and content are poor (see the comments above)”

With respect to the comments about “Section 3 English has to be edited, sentences be rephrased and more info be added”

According to the suggestions from the Reviewer, the English usage in the section was corrected, and more information was added.

With respect to the comments about “L 126-128 add refs regarding ‘These microorganisms can favor the growth and vitality of bivalves”

According to the suggestions from the Reviewer, one reference supporting the statement was added, in concrete:

Abasolo-Pacheco, F.; Campa-Córdova, A.I., Mazón-Suástegui, J.M.; Tovar-Ramírez, D.; Araya, R.; Saucedo, P.E. Enhancing growth and resistance to Vibrio alginolyticus disease in catarina scallop (Argopecten ventricosus) with Bacillus and Lactobacillus probiotic strains during early development. Aquaculture Research 2017, 48, 4597-4607.

With respect to the comments about “L 137-139 ‘it were isolated’ please remove and rephrase”

According to the suggestions from the Reviewer, the phrase was corrected

With respect to the comments about “Table 2 has to be enriched with more refs”

According to the suggestions from the Reviewer, Table 2 and the text corresponding to this Table was enriched with more references. Although there are large quantity of work published regarding the utilization of probiotics is the protection of bivalves against infections, we did not found more published work regarding use of probiotics in the improvement of the depuration process, that is our aims. However, other biological methods whose were not included in the original version (such use of antimicrobial peptides were included in the revised version of the manuscript.

With respect to the comments about “L 164-165 too many words, please make it simple”

According to the suggestions from the Reviewer, the phrase was corrected

With respect to the comments about “L 221-222 rephrase and remove the ‘above all’”

According to the suggestions from the Reviewer, the phrase was corrected

With respect to the comments about “L 223 Bivalves accumulate heavy metals...”

According to the suggestions from the Reviewer, the phrase was corrected

With respect to the comments about “L 232-233 rephrase”

According to the suggestions from the Reviewer, the phrase was corrected

With respect to the comments about “Sections 5 – 6 same as above (check English, rephrase in most cases, check for more info, etc)”

According to the suggestions from the Reviewer, English usage was corrected.

With respect to the comments about “Conclusions; L 321 remove ‘As final conclusion’ and The section has to be rephrased too.”

According to the suggestions from the Reviewer, “As a final conclusion” was deleted and the section was rephrased

Reviewer 2 Report

The paper is reasonably well written but should probably be read over by a native English speaker. The review seems fairly comprehensive but it does not say much about viruses

Line 18: Produce?

line 53: change "affect these species" to "contaminate shellfish"

Line 57: change to read "a particularly important point from a food safety standpoint is that the feeding process of bivalve mollusks is by water filtration."

Line 59: change to read "accumulate pathogenic bacteria, viruses, toxins, and chemical pollutants in their tissues that can pose a risk to public health"

Author Response

With respect to the comments from the Reviewer 2:

With respect to the comments about “The paper is reasonably well written but should probably be read over by a native English speaker”.

The authors want to thank the constructive comments from the reviewer. In fact, in none of the authors of the manuscript in a native English speaker and consequently it is normal that some English grammar mistakes are present. To avoid this problem, the authors used MDPI English editing service, that ensure a correct spelling in the revised version of the manuscript.

With respect to the comments about “The review seems fairly comprehensive, but it does not say much about viruses”

Thank you for your comment. Unfortunately, nowadays there are not many options for getting an effective elimination of viral contamination of bivalve mollusks prior to consumption, at least without changing the desired sensory characteristics of mollusks. For this reason, it was included short information about viruses in the original version of the manuscript.

Accordingly with the comments from the Reviewer, for the revised version of the manuscript it were carefully reviewer three new references related to viral contamination and decontamination in mollusks, and the most relevant information contained was used to complement the information about viruses throughout the revised version of the manuscript.

In concrete:

Razafimahefa, R.M.; Ludwig-Begall, L.F.; Thiry, E. Cockles and mussels, alive, alive, oh*-The role of bivalve molluscs as transmission vehicles for human norovirus infections. Transbound. Emerg. Dis. 2019, 1, 17.

Hodgson, K.R.; Torok, V.A.; Turnbull, A.R. Bacteriophages as enteric viral infections in bivalve mollusk management. Food Microbiol. 2017, 65, 284-293.

Naïma, B.M.H. Human enteric viruses in bivalve molluscs: Contamination and detection. Int. J. Sci. Technol. 2015, 4, 6.

With respect to the comments about “Line 18: Produce?”

According to the suggestions from the Reviewer, the sentence was modified.

With respect to the comments about “line 53: change "affect these species" to "contaminate shellfish"”

According to the suggestions from the Reviewer, it was modified.

With respect to the comments about “Line 57: change to read "a particularly important point from a food safety standpoint is that the feeding process of bivalve mollusks is by water filtration."

According to the suggestions from the Reviewer, it was modified.

With respect to the comments about “Line 59: change to read "accumulate pathogenic bacteria, viruses, toxins, and chemical pollutants in their tissues that can pose a risk to public health".

According to the suggestions from the Reviewer, it was modified.

Reviewer 3 Report

Summary:

This manuscript describes various techniques used to help remove contaminants and toxins from bivalve shellfish. Shellfish contamination is a significant problem causing risks to consumer health and generating great cost to the shellfish industry. No technique is successful at removing every toxin and methods are continually improving as technology evolves, so it is convenient to summarize the available techniques in one place.

Comments:

The authors do a good job with the scope of techniques compiled and the references used to support them, and this work will be valuable to the shellfish community. However, some techniques are described in much more detail than others. For example, Section 5 beginning on p. 8 is a very well written section of available chemical methods, with substantive discussions and details of the methods referenced. Section 3 is not as well-written; although it contains a thorough list of methods it is not as fully fleshed-out as Section 5 and is therefore less useful. The manuscript has the feel as though two different people wrote it, and it is lacking consistency throughout.

The writing style of the authors is awkward in many places, with complicated phrasing and inappropriate terms. This makes understanding the method descriptions difficult. Some examples: p. 1 line 27, p. 2 line 67, p 2 lines 90-93, p. 3 lines 121-122, p. 4 lines 137-138, p. 6 line 168, p. 7 lines 213-215. In the heading sections of Tables 2, 3, and 4 the term should be “species.” In Table 4, “quelant” should be replaced with “chelating.” In addition, the authors use the term “contrariwise” in many places. This is not a commonly used English term and should be avoided. It is recommended that the manuscript be reviewed again for English before resubmission.

Author Response

With respect to the comments from the Reviewer 3:

With respect to the comments about “This manuscript describes various techniques used to help remove contaminants and toxins from bivalve shellfish. Shellfish contamination is a significant problem causing risks to consumer health and generating great cost to the shellfish industry. No technique is successful at removing every toxin and methods are continually improving as technology evolves, so it is convenient to summarize the available techniques in one place.”

The authors would like to thank the reviewers for its constructive comments.

With respect to the comments about “The authors do a good job with the scope of techniques compiled and the references used to support them, and this work will be valuable to the shellfish community.”

The authors would like to thank the reviewers for its constructive comments.

With respect to the comments about “However, some techniques are described in much more detail than others. For example, Section 5 beginning on p. 8 is a very well written section of available chemical methods, with substantive discussions and details of the methods referenced.”

The authors would like to thank the Reviewer for its constructive comments.

With respect to the comments about “Section 3 is not as well-written; although it contains a thorough list of methods it is not as fully fleshed-out as Section 5 and is therefore less useful. The manuscript has the feel as though two different people wrote it, and it is lacking consistency throughout.”

The authors would like to thank the Reviewer for its constructive comments. In fact, as was described in the Author contributions sections, the manuscript was written by two different people, what has been perceived by the reviewer. The subsequent review and editing work should have corrected these differences, but it has not done so satisfactorily. In the new version of the manuscript, the length of section 3 has been increased, and the writing style has been corrected.

With respect to the comments about “The writing style of the authors is awkward in many places, with complicated phrasing and inappropriate terms. This makes understanding the method descriptions difficult. Some examples: p. 1 line 27, p. 2 line 67, p 2 lines 90-93, p. 3 lines 121-122, p. 4 lines 137-138, p. 6 line 168, p. 7 lines 213-215.”

According to the suggestions from the Reviewer, the cited paragraphs were rewritten and the English spelling of whole manuscript was reviewed by MDPI edition services.

With respect to the comments about “In the heading sections of Tables 2, 3, and 4 the term should be “species.””

According to the suggestions from the Reviewer, the term “specie” was changed to “species”.

With respect to the comments about “In Table 4, “quelant” should be replaced with “chelating.”

According to the suggestions from the Reviewer, the term “quelant” was changed to “chelating”.

With respect to the comments about “In addition, the authors use the term “contrariwise” in many places. This is not a commonly used English term and should be avoided.”

According to the suggestions from the Reviewer, the word “Contrariwise” was changed to other most accorded terms.

With respect to the comments about “It is recommended that the manuscript be reviewed again for English before resubmission.”

According to the suggestions from the Reviewer, the English spelling of whole manuscript was reviewed by MDPI edition services.

Round 2

Reviewer 1 Report

Authors have addressed the comments. However, English should be edited by a native speaker in order to improve the manuscript.